# Capabilities of Grazing Incidence X-ray Diffraction in the Investigation of Amorphous Mixed Oxides with Variable Composition

**DOI:** 10.3390/ma15062144

**Published:** 2022-03-15

**Authors:** Elisabetta Achilli, Filippo Annoni, Nicola Armani, Maddalena Patrini, Marina Cornelli, Leonardo Celada, Melanie Micali, Antonio Terrasi, Paolo Ghigna, Gianluca Timò

**Affiliations:** 1Materials and Generation Technologies Department (TGM), Ricerca sul Sistema Energetico S.p.A.—RSE, Strada Torre della Razza, le Mose, 29100 Piacenza, Italy; filippo.annoni@rse-web.it (F.A.); nicola.armani@rse-web.it (N.A.); marina.cornelli@rse-web.it (M.C.); gianluca.timo@rse-web.it (G.T.); 2Department of Physics, University of Pavia, Via Bassi 6, 27100 Pavia, Italy; maddalena.patrini@unipv.it; 3Department of Physics, University of Parma, Via Università 12, 43121 Parma, Italy; leonardo.celada@studenti.unipr.it; 4Department of Physics and Astronomy “Ettore Majorana”, University of Catania & IMM-CNR Sede Catania, Via S. Sofia 64, 95123 Catania, Italy; melanie.micali@ct.infn.it (M.M.); antonio.terrasi@ct.infn.it (A.T.); 5Department of Chemistry, University of Pavia, Via Taramelli 12, 27100 Pavia, Italy; paolo.ghigna@unipv.it

**Keywords:** GIXRD, amorphous oxides, mixed oxides, thin films, XRR, RBS, ellipsometry

## Abstract

X-ray Diffraction has been fully exploited as a probe to investigate crystalline materials. However, very little research has been carried out to unveil its potentialities towards amorphous materials. In this work, we demonstrated the capabilities of Grazing Incidence X-ray Diffraction (GIXRD) as a simple and fast tool to obtain quantitative information about the composition of amorphous mixed oxides. In particular, we evidenced that low angle scattering features, associated with local structure parameters, show a significant trend as a function of the oxide composition. This evolution can be quantified by interpolating GIXRD data with a linear combination of basic analytical functions, making it possible to build up GIXRD peak-sample composition calibration curves. As a case study, the present method was demonstrated on Ta_2_O_5_–SiO_2_ amorphous films deposited by RF-magnetron sputtering. GIXRD results were validated by independent measurement of the oxide composition using Rutherford Backscattering Spectrometry (RBS). These materials are attracting interest in different industrial sectors and, in particular, in photovoltaics as anti-reflection coatings. Eventually, the optical properties measured by spectroscopic ellipsometry were correlated to the chemical composition of the film. The obtained results highlighted not only a correlation between diffraction features and the composition of amorphous films but also revealed a simple and fast strategy to characterize amorphous thin oxides of industrial interest.

## 1. Introduction

Amorphous layers obtained from the mixture of two traditionally well-known oxides are becoming widespread and crucial both in fundamental research and in many industrial fields. The main reason behind this recent interest lies in the possibility of tuning their electronic, microstructural and optical properties by varying their composition [1]. Their applications span from energy and environment [2] to domestic and industrial sectors [3]. They show excellent potentialities as gas sensors [4,5], electro-catalysts for water splitting reactions [6] and protective layers with suitable mechanical and thermal properties [7]. Moreover, they are investigated as anti-reflection coatings (ARCs) for tackling reflection and absorption phenomena in solar cells [8,9,10].

For the development of such mixed oxide materials, the characterization of their physical, chemical and electro-optical properties is of key importance. X-ray Reflectivity (XRR) is undoubtedly the most reliable and effective tool to quantitatively investigate thickness, roughness and density and can be carried out by a laboratory diffractometer. Even though density depends on atomic composition, it cannot be directly related to that since other features such as porosity and compactness are involved. From a chemical point of view, we should consider (i) composition in terms of element abundance and (ii) structure in terms of coordination geometry and phase. As for the former, X-ray Fluorescence, Secondary Ion Mass Spectrometry and Rutherford Backscattering Spectrometry (RBS) are commonly employed. For the latter, local probes such as X-ray Photoelectron Spectroscopy or X-ray Absorptn Spectroscopy are the most widespread. All the techniques listed above require specific and additional instrumentations different from a conventional laboratory diffractometer.

In this work, we demonstrated the potentialities of Grazing Incidence X-ray Diffraction (GIXRD) as a fast and easy tool for quantitative chemical characterization of amorphous mixed compounds. With respect to traditional XRD, GIXRD is the more appropriate tool for investigating thin films (around tens of nanometres thick) as a grazing X-ray beam is more sensitive to the near-surface region. It is well established that amorphous materials give rise to “broadened structures” at low angles. These features are associated with the local chemical environment, and the greater the atomic weight is (and therefore the scattering cross-section), the higher the GIXRD intensity is [11]. Thus far, diffractograms of amorphous samples were employed to establish their non-crystalline nature; however, no exploitation of this technique has been carried out to obtain quantitative information concerning the short-range chemical environment. 

Hereafter we performed a GIXRD analysis focusing on mixed layers, obtained by Radio Frequency (RF)-magnetron sputtering, constituted of SiO_2_ and Ta_2_O_5_ oxide, in different percentages. GIXRD curves were first shown to have a trend with composition. Rutherford Backscattering Spectrometry (RBS) and Spectroscopic ellipsometry analysis were also reported as further and independent validation of the GIXRD data interpretation. Then, X-ray Reflectivity (XRR) was carried out in order to investigate samples from a physical point of view by the same instrumental setup. Finally, spectroscopic ellipsometry was employed to correlate the optical properties to GIXRD analysis, demonstrating that GIXRD can constitute a fast and powerful characterization method for investigating amorphous mixed compounds. In particular, our interest lies in developing anti-reflection coatings for multi-junction solar cells [12,13,14].

## 2. Materials and Methods

RF Magnetron sputtering was employed to deposit Ta_2_O_5_ and SiO_2_ thin films and their mixtures, starting from nominally stoichiometric oxide targets (K.J. Lesker Ltd., Dresden, Germany). The deposition chamber was equipped with two 3” sputtering cathodes (Kenosistc Srl, Binasco (MI), Italy), powered by independent RF generators. The oxide deposition took place over a Silicon (100) substrate installed on a water-cooled (20 °C) rotating-sample holder (20 rpm). The target-substrate distance was kept fixed at 18 cm for all the samples, and the process gas was Ar (6N). Before deposition, targets were prepared by polishing, and a conditioning procedure was carried out by increasing the sputtering power by means of ramps and then by lowering the chamber pressure to the desired values. No intentional heating was applied during and after deposition. In the beginning, pure Ta_2_O_5_ and pure SiO_2_ were deposited by RF sputtering at a power of 110 and 200 W, respectively, varying argon pressure from 1.0 × 10^−3^ to 1.0 × 10^−2^ mbar in order to identify the conditions at which deposition rate and material density were maximized. At this stage, only the cathode hosting the investigated oxide was turned on. The deposition time was varied to obtain films with thickness in the range of 30–100 nm, suitable for XRR measurements. Then, oxides mixtures were deposited by a co-sputtering method, in which the two independent sputtering cathodes were used simultaneously. Firstly, at an argon pressure of 5.0 × 10^−3^ mbar, a deposition rate calibration was carried out for each material as a function of sputtering power (ranging from 75 to 200 W for SiO_2_ and from 40 to 140 W for Ta_2_O_5_). During these calibration runs, both cathodes were turned on, but the shutter of the material not under investigation was kept closed to prevent any deposition. The volumetric Ta_2_O_5_/SiO_2_ composition was fixed by the Ta_2_O_5_/SiO_2_ thickness ratio, controlled by varying the sputtering power applied to the targets. The compositional range explored varied from pure Ta_2_O_5_ to pure SiO_2_. Sputtering time was kept fixed at 25 min to obtain a film thickness of about 50 nm. 

In order to check for the actual composition of the series of mixed samples (numbered from #1 to #5 (see Appendix A)), RBS analysis was applied. Specifically, RBS was employed to quantify the doses of the chemical elements (cm^−2^) and then to calculate the composition of the films. Measurements were carried out with a 2.0 MeV He^+^ ion beam, and SimNra software was employed to simulate the RBS spectra for the quantitative analysis. Spectra of SiO_2_ and Ta_2_O_5_/SiO_2_ thin films mixtures were collected in glancing-angle configuration with an incident angle α = 40° and backscattered ions detected at θ = 165°, while the spectrum of Ta_2_O_5_ thin film was collected at normal incidence α = 0. The glancing angle was used to increase the accuracy of the measurement in the mixed oxide samples, where the signal of the Si in the film had to be separated from that of the substrate.

Spectroscopic ellipsometry (SE) data were measured by a VASE spectrometer by J.A. Woollam Inc. in the spectral range from 0.5 to 5 eV at different angles of incidence from 65° to 75°. Experimental spectra were analyzed through the dedicated WVASE32^®^ software and database.

X-ray investigation was carried out by means of a Bruker D8 Advance diffractometer working at Cu *Kα* radiation. Grazing Incidence X-ray Diffraction measurements were acquired by maintaining a fixed incidence angle of 1.3° and slit widths of either 2.0 mm or 0.6 mm for both input and output beams to the detector. X-ray Reflectivity analysis was carried out applying 0.2 mm slits both in the entrance and before the detector. The polycrystalline Ta_2_O_5_ standard diffractogram was acquired in Bragg Brentano geometry on a 99.99% pure sputtering target supplied by Kurt J. Lesker Company (Dresden, Germany). Data visualization and analysis were carried out by Leptos software [15], delivered by Bruker corporation, and by standard calculation and fitting software. 

## 3. Results and Discussion

The RBS spectra report the signal due to He^+^ ions backscattered by different chemical elements at different depths, including the substrate. Once the spectrum was recorded, it was compared with a simulation obtained by the SimNra commercial software. In order to perform the spectrum analysis, the operator sets all the physical parameters of the measurement (e.g., incident ion type and energy), defines the scattering geometry with the incidence and backscattered angles and enters the energy calibration of the experiment. The sample was simulated with a layer structure (film-substrate), each layer being defined by the atomic percentage concentration of the different chemical elements and their thickness. The atomic doses of the chemical elements detected (Si, Ta, O) were then converted in volumetric fractions of SiO_2_–Ta_2_O_5_ for the mixed oxides. As an example, the RBS spectrum of sample #3 with an intermediate composition is reported in Figure 1 along with its SimNra simulation. The final output of SimNra was obtained with 38% Ta_2_O_5_ and 62% SiO_2_ volumetric fractions. The RBS spectra and simulations of the other samples are reported in Appendix A.

The preliminary GIXRD and XRR investigation carried out on layers constituted of pure Ta_2_O_5_ (prepared at different Ar pressures) is shown in Appendix A. 

Hereafter we report the investigation on mixed samples, in which Ta_2_O_5_ was combined with different percentages of SiO_2_. Figure 2 shows the series of GIXRD diffractograms of Ta_2_O_5_/SiO_2_ mixed layers in the *2θ* angular range between 5 and 42°, acquired in two experimental conditions: (a) with 0.6 mm and (b) with 2.0 mm slit width for both input and output beams.

As expected, the opening of the reduced slit determines a decrease in signal intensity, but it results in an overall better signal-to-noise ratio. This is due to the smaller slit opening that reduces the diffuse scattering contribution [16]. As a consequence, the abatement in the background signal is much more pronounced with respect to the reduction in scattering contribution; therefore, the relevant signal appears more intensive. At first sight, it is evident that the larger is the tantalum content, the higher the intensity of the signal. Quite interestingly, it can be pointed out that the “hump” located below 30° shifts towards larger angles by increasing Ta_2_O_5_ content, while the one located at nearly 35°, when present, appears fixed. Then, to quantify the shift of the first maximum position and to investigate its evolution with composition, the profile features were analyzed through a best-fit procedure with variable parameters to GIXRD curves, excluding SiO_2_ 100%. This sample was not considered for the low marked GIXRD features and then the too large uncertainty on fitting parameters. The fitting was carried out with a linear combination of a Lorentzian function (to simulate the background signal) and one or two Gaussian peaks modeling the humps. The choice of a Lorentzian line-shape to model the background trend was based on previous evaluation of best-fits involving different functions (*1/ax*, exponential and Lorentzian) reported in Appendix A, while the choice of Gaussian functions to simulate low angle scattering features was due to both instrumental and physical aspects. It is well known that typical Bragg peaks arising in a conventional laboratory XRD experiment are well represented by Voigt functions. Both instrumental setup and sample crystallinity affect peak broadening. Specifically, Lorentzian contribution is mainly due to instrumental setup, whilst sample crystallinity mainly controls the Gaussian part. When dealing with materials with low crystallinity degree, the Gaussian part becomes dominant, and in amorphous samples, as in the present case, peaks broaden into hump-like structures, and peak fitting can be safely carried out with Gaussians neglecting the Lorentzian part [17].

The simulated curves here obtained are given by:(1)$$y_{calc.}=\left[\frac{2A_{L}}{\pi }\;\frac{\sigma_{L}}{4{\left(x-x_{0L}\right)}^{2\;}+{\left(\sigma_{L}\right)}^{2}}\right]+\left[\frac{A_{1}}{\sigma_{1}\sqrt{2\pi }}e^{\frac{-{\left(x-x_{01}\right)}^{2}}{2\sigma_{1}^{2}}}\right]+\left[\frac{A_{2}}{\sigma_{2}\sqrt{2\pi }}e^{\frac{-{\left(x-x_{02}\right)}^{2}}{2\sigma_{2}^{2}}}\right]$$
where ***A_i_*** is the integrated area, ***x_oi_*** is the maximum peak position, and ***σ_i_*** is the standard deviation of Lorentzian (***L***) and Gaussian Functions (***i***
*=* 1,2). The first two terms of the model were best-fit to all diffractograms reported in Figure 2. For samples with Ta_2_O_5_ content greater than 30%, the second Gaussian function was added in order to simulate the observed “double hump” structure better. As an example, Figure 3 illustrates the fit results obtained for a sample with 74% Ta_2_O_5_ and 26% SiO_2_ (a) and for a sample of 6% Ta_2_O_5_ and 94% SiO_2_ (b). 

Fit results concerning 1st and 2nd Gaussian peak angular positions are listed in Table 1 for all samples and both experimental configurations. The excellent fit quality is indicated by the R-factor (which describes the difference between the experimental observations and the fitting curves) lower than 0.001 in all cases. 

In order to test the reliability of this approach, for each mixed oxide composition, the GIXRD curve was acquired at different incidence angles, between 0.6 and 1.3°. Furthermore, measurements were carried out in both experimental configurations. The analysis results show that best-fit parameter values do not vary over the fitting error, both as a function of different incidence angles and by using different slit widths. By excluding measurement artifacts, it seems reasonable to assume that the evolution of the GIXRD pattern can only be ascribed to the chemical nature of the samples. The trend of the Gaussian maximum positions with respect to the oxide composition is illustrated in Figure 4 for both 0.6 mm (a) and 2.0 mm (b) slit openings. 

As expected from a qualitative observation of GIXRD profiles, only the first Gaussian peak shows a relevant shift in the maximum, whereas the second one, when present, always remains at the same angular value as a function of the composition. For the series of GIXRD curves acquired, the first Gaussian maxima are well interpolated by parabolic functions. Parameters are similar for the two series of data, and R squared values are greater than 0.95 in both cases. We underline that the polynomial function obtained to interpolate these data can be considered as a calibration curve in order to obtain quantitative information concerning the chemical composition of Ta_2_O_5_/SiO_2_ mixed oxide layers by only considering their GIXRD diffractograms. 

Gaussian maxima significance can be related to short-range structural parameters. In fact, amorphous phases can be considered as phases in which crystallite dimensions are limited to a few unit cells (that is, a few tenths of angstroms). Scattering profiles arise from the broadening of Bragg peaks and are generated by the radial distribution function. Thence, Gaussian maxima obtained can be associated with the local chemical environment in terms of interatomic distances and angles. To discuss our specific case, in Figure 5, a diffractogram of crystalline Ta_2_O_5_ (obtained from sputtering target material) is compared to the GIXRD curve obtained in correspondence of amorphous Ta_2_O_5_ pure oxide film. 

All diffraction peaks are indexed by means of the *Pccm* orthorhombic phase, for which the most intense reflections are 002 (*ca.* 22.8°), 110 (*ca.* 28.2°) and 112 (*ca.* 36.5°). By comparing the two GIXRD patterns, the first hump in the diffractogram of the amorphous sample can be associated with the broadening of 002 and 110 reflections, whereas the main contribution to the second peak can come from the 112 reflections. If this association is correct, it is reasonable to assume that the variation in the relative composition of Ta_2_O_5_/SiO_2_ mixed oxides mainly affects local structure parameters along the horizontal sides of the unit cell. A similar discussion for pure amorphous SiO_2_ is not relevant for the present investigation since its GIXRD diffractogram does not show low-angle scattering features as a consequence of the lower weight. Therefore, the evolution in the diffraction patterns can be described by considering the Ta_2_O_5_ structure. A similar methodology can, in principle, be extended to a wide variety of amorphous mixed materials, provided the scattering cross-section of the elements is sufficiently large to generate well distinguishable scattering features.

As far as the physical investigation is concerned, film density, thickness and roughness of the mixed samples were accurately obtained with XRR investigation. In Figure 6, we can observe XRR curves recorded up to 7° in *2θ*. 

Three are the main features associated with an XRR curve. (i) The position of the critical angle, associated with the total reflection conditions, is an experimental characteristic directly associated with the density of the material under investigation. Density value confirm the composition of the samples obtained by RBS analysis. (ii) The frequency of Kiessig oscillations, generated by the thickness of the deposited films, provides a suitable starting parameter for the subsequent fitting procedure. We obtained a minimum thickness in correspondence of Ta_2_O_5_ 100% (47.7 nm) and a maximum one for sample Ta_2_O_5_ 74%-SiO_2_ 26% (59.9 nm). The presence of a low-frequency component in the XRR curve of sample Ta_2_O_5_ 38%-SiO_2_ 62% can be reasonably ascribed to a slightly thicker native oxide layer formed at the interface just before the layer’s growth. Its density is sufficient to generate an appreciable contrast. (iii) Signal decay is related to layer roughness and quality of the interfaces. The best fit parameter values are listed in Table 2. 

Spectroscopic ellipsometry (SE) was eventually coupled to the GIXRD measurements to correlate the film’s optical properties to the chemical sample composition. In the analysis of the experimental SE spectra, we adopted a tri-layer model, plus an additional overlayer to account for an rms surface roughness effect. The three layers of the sample structure were the Si substrate, its native silica oxide and the sputtered oxide film. This allowed, through WVASE32^®^ software, (i) to determine the pseudo-dielectric function spectra of the film by best-fitting simulated spectra to the experimental ones, (ii) verify the actual film thickness values (the actual thickness values determined by SE are typically within 5% to the XRR ones) and (iii) check the optical quality and uniformity of the films. 

For the best-fit analysis, we modeled the dielectric function of the oxide films with Tauc–Lorentz and Gaussian oscillators (Kramers–Kronig consistency is then guaranteed), obtaining a very good agreement between experimental and fitted spectra. The real part of the complex refractive index (see Figure 7) of the end-member Ta_2_O_5_ and SiO_2_ films was first derived and compared to the literature data; then, mixed oxide films were analyzed. In addition, variable parameters of the best fit were the thickness for the film itself and also for the roughness overlayer.

The overall refractive index spectra demonstrate both the modulation of the optical response with varying film composition and the transparency region (low values of the extinction coefficient) through the whole NIR-vis (Near InfraRed/visible) range (not shown here). In a typical anti-reflection coating application, this leads to the possibility to gradually change the refractive index in order to decrease the reflected radiation amount.

Figure 8 shows a correlation graph of GIXRD and ellipsometry results. For each composition, the refractive index at 630 nm is plotted against the first Gaussian maximum position. 

Results are shown for measurements with 0.6 mm Slits characterized by a better signal-to-noise ratio, as discussed above. By performing a polynomial fit, it is easy to observe that data are very well interpolated by a second-order function, which can serve as a final calibration curve for the easy and fast association of GIXRD features (and therefore of the film composition) to optical properties. Furthermore, we can safely affirm that four is an adequate number of mixed samples to obtain a reliable and effective calibration function.

## 4. Conclusions

In this work, GIXRD was exploited as a tool for a quantitative investigation concerning amorphous mixed oxides composition. We demonstrated that the broad structures visible at low *2θ* angles, associated with short-range structure features, show a trend with composition. This trend was quantified through a peak-fit analysis, and the resulting interpolation functions can be employed as calibration curves to perform a quantitative determination of the film composition. We applied this characterization methodology to amorphous layers composed of Ta_2_O_5_ and SiO_2_ in different compositions, which, in turn, attracted interest as anti-reflection coatings for solar cell devices.

The actual composition of the samples employed for the calibration curve was determined by RBS, which gives an absolute measurement of the atom doses and, thus, of their percentages in the mixed oxides.

XRR analysis was subsequently performed to study physical characteristics in terms of density, thickness and roughness. Eventually, spectroscopic ellipsometry measurements were performed to correlate the film’s optical properties to the mixed oxide composition. This work evidences that it is possible to investigate the physical, chemical and optical properties of mixed oxides by coupling GIXRD with XRR. This characterization strategy provides an effective method to be applied in industrial scale application, as it involves fast measurements with conventional laboratory instruments.

## Figures and Tables

**Figure 1 materials-15-02144-f001:**
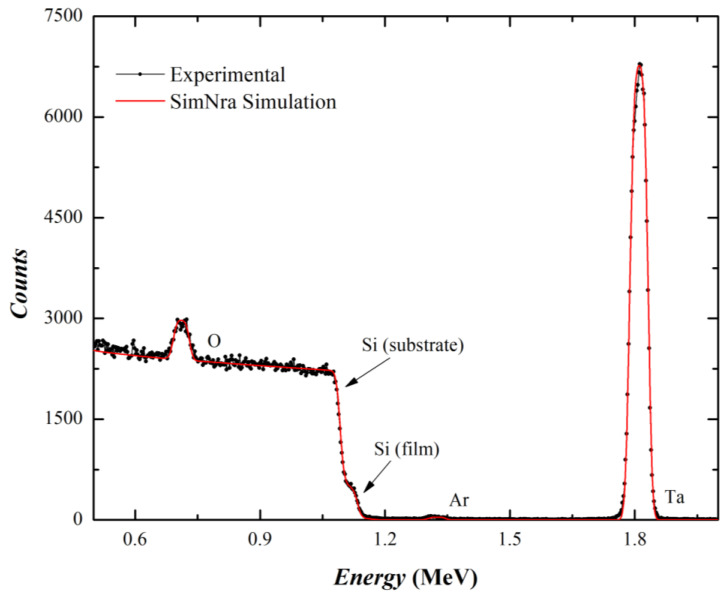
Experimental and simulated RBS spectra of the mixed oxide film–sample#3.

**Figure 2 materials-15-02144-f002:**
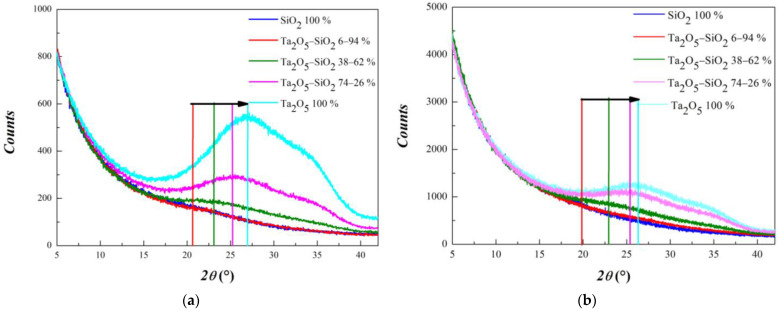
GIXRD curves obtained for Ta_2_O_5_/SiO_2_ mixed layers with different volumetric composition in the angular range 5–42° with 0.6 (**a**) and 2.0 mm slit (**b**). The colored segments are added in order to qualitatively show the evolution of the first “structure” characterizing the diffraction profiles.

**Figure 3 materials-15-02144-f003:**
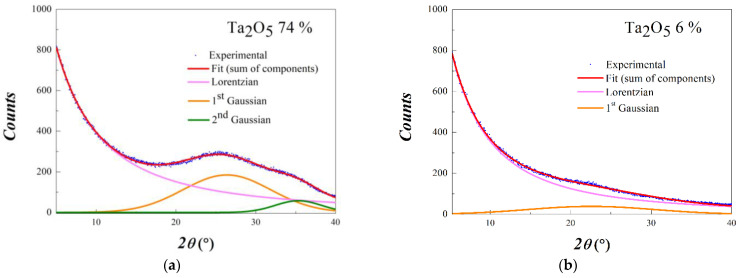
(**a**) Best-fit of GIXRD curve obtained for the sample with 74% Ta_2_O_5_ and 26% SiO_2_; (**b**) best-fit of GIXRD curve obtained for the sample with 6% Ta_2_O_5_ and 94% SiO_2_. Both curves were obtained by employing 0.6 mm slit widths for input and output beams. Experimental data are indicated by the blue dots, whereas the best-fit curve is the red line.

**Figure 4 materials-15-02144-f004:**
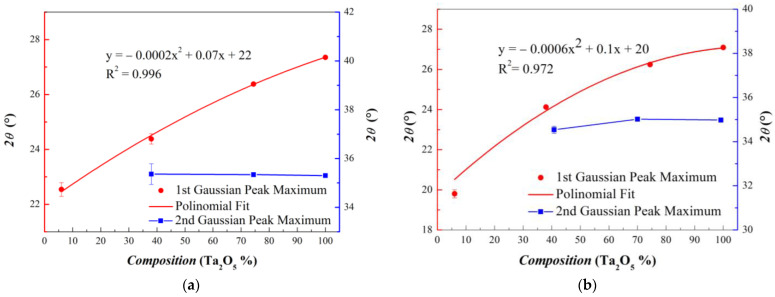
Trends of the Gaussian maxima from the best fit to GIXRD series acquired with Slit widths equal to 0.6 mm (**a**) and 2.0 mm (**b**).

**Figure 5 materials-15-02144-f005:**
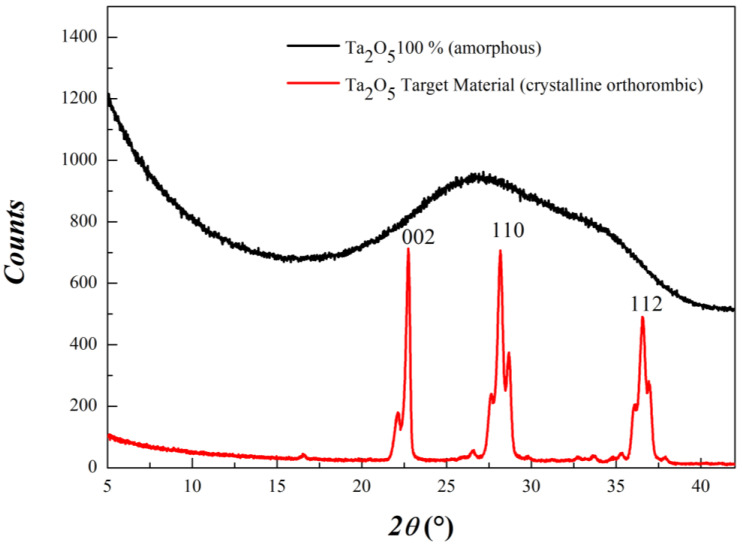
Comparison between the diffractograms of *Pccm* orthorhombic crystalline Ta_2_O_5_ (**red line**) and amorphous Ta_2_O_5_ (**black line**).

**Figure 6 materials-15-02144-f006:**
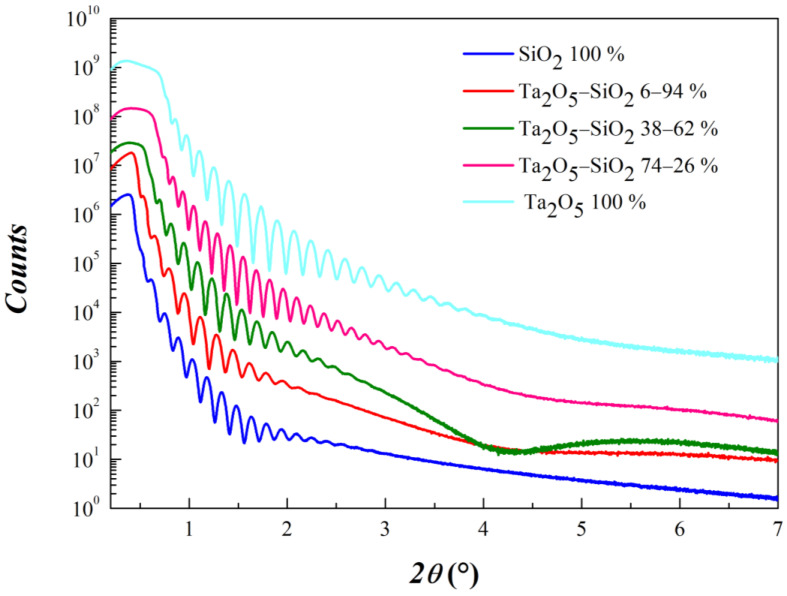
Comparison of XRR curves on Ta_2_O_5_/SiO_2_ mixed oxide samples with different compositions obtained at Ar pressure equal to 2.5 × 10^−3^ mbar. Ar flow was kept at 3.9 sccm.

**Figure 7 materials-15-02144-f007:**
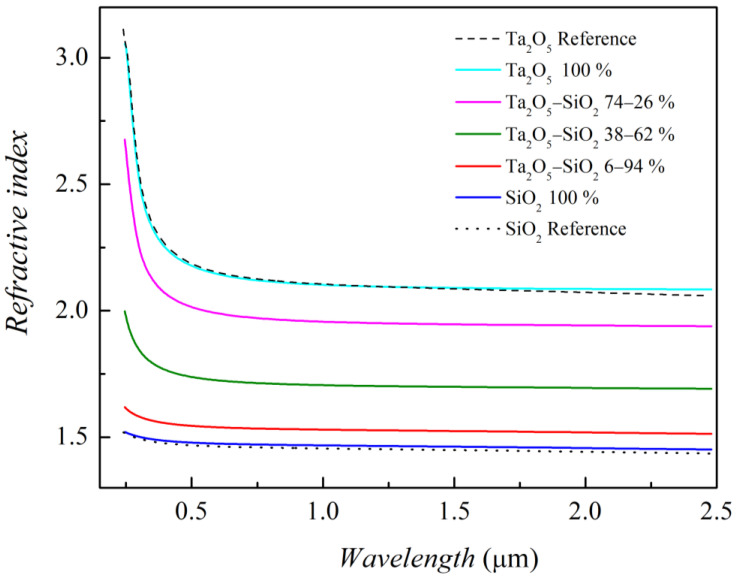
Refractive index spectra on Ta_2_O_5_/SiO_2_ mixed oxide samples. Ta_2_O_5_ and SiO_2_ reference curves were obtained from WVASE32^®^ Database by J.A. Woollam Co. Inc.

**Figure 8 materials-15-02144-f008:**
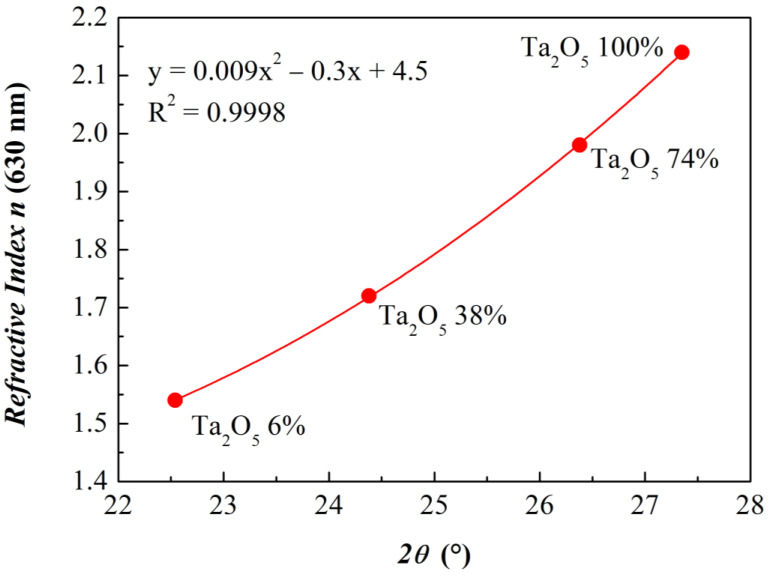
Trend of refractive indexes at 630 nm with respect to the first Gaussian maximum positions from the best fit to GIXRD series (shown in Figure 2a). Results are from acquisitions with Slit widths equal to 0.6 mm.

**Table 1 materials-15-02144-t001:** Best-fit peak angular positions of GIXRD curves shown in Figure 2 referred to 0.6 (**a**) and 2.0 mm slit widths (**b**).

**(a) Film (Volumetric Percentage)**	**1st Gaussian Peak (°)**	**2nd Gaussian Peak (°)**
94% SiO_2_ 6% Ta_2_O_5_	22.5 (2)	/
62% SiO_2_ 38% Ta_2_O_5_	24.4 (2)	35.4 (4)
26% SiO_2_ 74% Ta_2_O_5_	26.38 (6)	35.34 (9)
100% Ta_2_O_5_	27.35 (6)	35.30 (8)
**(b) Film (Volumetric Percentage)**	**1st Gaussian Peak (°)**	**2nd Gaussian Peak (°)**
94% SiO_2_ 6% Ta_2_O_5_	19.8 (2)	/
62% SiO_2_ 38% Ta_2_O_5_	24.12 (6)	34.5 (2)
26% SiO_2_ 74% Ta_2_O_5_	26.24 (4)	35.02 (6)
100% Ta_2_O_5_	27.09 (3)	34.98 (4)

**Table 2 materials-15-02144-t002:** Best fit parameter values refer to XRR curves shown in Figure 6.

Film (Volumetric Composition)	Thickness (nm)	Density (g/cm^3^)	Roughness (nm)
100% SiO_2_	55.2 (5)	2.1 (1)	2.7 (1)
94% SiO_2_ 6% Ta_2_O_5_	48.5 (5)	3.2 (1)	2.4 (1)
62% SiO_2_ 38% Ta_2_O_5_	54.7 (5)	5.0 (1)	1.9 (1)
26% SiO_2_ 74% Ta_2_O_5_	59.9 (5)	6.7 (1)	1.6 (1)
100% Ta_2_O_5_	47.7 (5)	8.0 (1)	1.3 (1)

## Data Availability

The data that support the findings of this study are available within the article and from the corresponding author upon reasonable request.

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
