# Peer review of "Capabilities of Grazing Incidence X-ray Diffraction in the Investigation of Amorphous Mixed Oxides with Variable Composition"

_materials, 2022, doi:10.3390/ma15062144_

Round 1

Reviewer 1 Report

Comments on Manuscript “materials-1621242”

The manuscript “Capabilities of Grazing Incidence X-ray Diffraction in the Investigation of Amorphous Mixed Oxides with Variable Composition” has been carefully reviewed.

In this work, the authors used RF sputtering system with two cathodes to deposit different chemical composition of TaOx and SiOx mixed amorphous films on Si (100) substrates from Ta2O5 and SiO2 targets. Moreover, the TaOx and SiOx mixed films (TaxSiyOz films) were characterized by GIXRD. Furthermore, the authors claim that GIXRD can be served as a quick and easy tool for quantitatively characterizing amorphous thin films of mixed compounds.

In general, GIXRD is widely used to identify the crystalline structure of thin films, but not the chemical composition. According to the analysis principle of GIXRD, GIXRD is not suitable for analyzing the chemical composition of thin films. In fact, there are many techniques widely used to analyze the chemical composition of thin films, including XPS, AES, SIMIS, RBS, etc., but GIXRD is not.

If the authors want to claim that “GIXRD can be used to quantitatively characterize the chemical composition of thin films”, more evidence is required.

Reviewer 2 Report

In this article the authors have investigated the application of XRD for amorphous materials nd have come to the conclusion that a quantitative analysis of the composition is possible using grazing incidence x-ray diffraction. In the first look the article is very interesting, The reviewer has few questions :-

  1. Please illustrate these lines - 150-153, as to why those functions were selected.
  2. Since this is a new technique, what limitations have the researchers faced in measuring amorphous materials  ?
  3. Do the researchers claim that this technique can be widely used for a variety of amorphous materials ?

The correlation between the GIXRD and the Ellipsometry technique is well taken.

The reviewer therefore recommends publishing of this article.

Reviewer 3 Report

In the present paper entitled “Capabilities of Grazing Incidence X-ray Diffraction in the Investigation of Amorphous Mixed Oxides with Variable Composition” There discusses the potentialities of Grazing Incidence X-ray Diffraction (GIXRD) as a fast and easy tool for a quantitative chemical characterization of amorphous mixed compounds. By performing a GIXRD analysis focusing on mixed layers constituted of SiO2 and Ta2O5 oxide, which obtained by Radio Frequency (RF)-magnetron sputtering, in different percentages, The results firstly found GIXRD curves having a trend with composition. Spectroscopic ellipsometry is employed to correlate the optical properties to GIXRD analysis, demonstrating that GIXRD can constitute a fast and powerful characterization method for investigating amorphous mixed compounds. In my opinion, the present work is quite interesting, but there are some suggestions need to pay attention:

1.I think you should extra text the sample “Ta2O5-SiO2 50-50%”.

  1. Although it shows a linear trend, but there would be better to illustrate its repeatability.
  2. The picture in the article is not very clear, it’s better to increase the resolution of the picture.
  3. The capitalization of letters in the title of the reference should be consistent.

Round 2

Reviewer 1 Report

In this revised version, the authors have added relevant experimental data in the main text and supplementary materials, and carefully revised the manuscript, greatly improving the quality of the article. Therefore I recommend accepting this paper in the present form.